# *Inula britannica* Inhibits Adipogenesis of 3T3-L1 Preadipocytes via Modulation of Mitotic Clonal Expansion Involving ERK 1/2 and Akt Signaling Pathways

**DOI:** 10.3390/nu12103037

**Published:** 2020-10-03

**Authors:** Hyung-Seok Yu, Won-Ju Kim, Won-Young Bae, Na-Kyoung Lee, Hyun-Dong Paik

**Affiliations:** Department of Food Science and Biotechnology of Animal Resources, Konkuk University, Seoul 05029, Korea; hyungseok_yu@naver.com (H.-S.Y.); jootopaz22@naver.com (W.-J.K.); won5483101@naver.com (W.-Y.B.); lnk11@konkuk.ac.kr (N.-K.L.)

**Keywords:** *Inula britannica*, anti-obesity, adipogenesis, lipogenesis, mitotic clonal expansion, ERK 1/2 signaling pathways, Akt signaling pathways

## Abstract

The flower of *Inula britannica* contains various phenolic compounds with prophylactic properties. This study aimed to determine the anti-adipogenic effect of an *I. britannica* flower aqueous extract (IAE) and its underlying mechanisms in the 3T3-L1 preadipocytes and to identify the phenolic compounds in the extract. Treatment with IAE inhibited the adipogenesis of 3T3-L1 preadipocytes by showing a dose-dependently suppressed intracellular lipid accumulation and significantly mitigated expression levels of lipogenesis- and adipogenesis-associated biomarkers including transcription factors. IAE exerted an anti-adipogenic effect through the modulation of the early phases of adipogenesis including mitotic clonal expansion (MCE). Treatment with IAE inhibited MCE by arresting the cell cycle at the G0/G1 phase and suppressing the activation of MCE-related transcription factors. Furthermore, IAE inhibited adipogenesis by regulating the extracellular signal-regulated kinase 1/2 and Akt signaling pathways. Protocatechuic acid, chlorogenic acid, kaempferol-3-*O*-glucoside, and 6-methoxyluteolin, which are reported to exhibit anti-adipogenic properties, were detected in IAE. Therefore, modulation of early phases of adipogenesis, especially MCE, is a key mechanism underlying the anti-adipogenic activity of IAE. In summary, the anti-obesity effects of IAE can be attributed to its phenolic compounds, and hence, IAE can be used for the development of anti-obesity products.

## 1. Introduction

Adipose tissue is crucially involved in various biological functions such as energy homeostasis, hormonal regulation, and metabolism by secreting the hormones, growth factors, and adipokines as well as function as an energy reservoir [1,2]. However, a chronic imbalance between the intake and consumption of energy promotes the aberrant growth of adipose tissue accompanying hyperplasia and/or hypertrophy of the adipocytes, consequently resulting in the development of obesity [3]. Obesity is a growing socioeconomic health concern as it is pathologically associated with development of the various degenerative disease such as type-2 diabetes mellitus, hypertension, dyslipidemia, and cardiovascular disease [4]. According to recent reports of the Centers for Disease Control and Prevention, prevalence of obesity in the United States of America has been increased over the past decade and it was estimated that approximately 20% of children and 40% of adults were obese in 2017–2018 [5]. Additionally, the increasing prevalence of obesity has led to an annual increase in extra medical expenditure for obesity and obesity-associated disorders as a socioeconomic burden [6]. Hence, considerable funds have been invested in the development of therapeutic strategies for obesity. Major strategies to mitigate the disease include decreasing appetite, enhancing energy expenditure, and inhibiting lipogenesis [7].

Adipogenesis, which is a multi-step process of adipocyte formation, involves commitment of pluripotent mesenchymal stem cells into preadipocytes and the differentiation of preadipocytes into mature adipocytes. The 3T3-L1 cell line, which is derived from the murine preadipocyte, is a well-established model for studying adipogenesis and can be differentiated into mature adipocytes under experimental conditions [8,9]. Adipogenesis in the 3T3-L1 preadipocytes comprises an early phase, an intermediate phase, and a terminal phase of differentiation. During adipogenesis, the preadipocytes are first growth-arrested, thereafter treated with hormonal inducers to initiate the early phases of differentiation, which includes synchronous re-entry into the cell cycle and mitotic clonal expansion (MCE). Subsequently, the cell cycle is terminated to undergo terminal phases of differentiation, which is associated with sequential changes in gene expression [10,11,12]. The MCE is a prerequisite for the differentiation of 3T3-L1 preadipocytes, underscoring the correlation between the cell cycle regulation and the differentiation in this adipogenesis model system [13,14]. Additionally, the activation of extracellular signal-regulated kinase (ERK) 1/2 and Akt signaling cascades is responsible for the MCE, as these signaling pathways mediate cell survival and cell cycle progression, consequently promoting adipogenesis [15,16].

Adipogenesis is modulated by multiple transcription factors such as signal transducer and activator of transcription-3 (STAT3), CCAAT/enhancer-binding proteins (C/EBPs), and peroxisome proliferator-activator receptor (PPAR)-γ [17]. During the early phase of differentiation, STAT3 and C/EBP-β are expressed in response to hormonal stimulation, continuously acquiring the DNA-binding activity to undergo MCE, and finally triggering the activation of C/EBP-α and PPAR-γ, the major transcription factors in terminal differentiation. The activation of C/EBP-α and PPAR-γ leads to the termination of MCE and modulation of the expression of genes involved in the differentiated phenotype [18]. In the adipocytes, de novo lipogenesis is markedly increased during terminal differentiation due to the upregulated expression of triacylglycerol-associated enzymes such as fatty acid-binding protein (FABP) 4, fatty acid synthase (FAS), perilipin, and stearoyl-CoA desaturase (SCD)-1. Particularly, perilipin is involved in the storage of lipids by coating lipid droplets and preventing lipolysis. This results in the accumulation of intracellular lipids, which promotes the expansion of the adipose tissue [7,12,14].

In response to increasing awareness of personal health, dietary polyphenols are garnering interest in the prevention and improvement of various disorders as an alternative medicine based on their biological functionalities [19]. Nutritional, clinical, and epidemiological studies have supported the evidence that appropriate dosage of dietary phenolic compounds improves human health by decreasing the risk and preventing the development of degenerative disorders with reduced adverse effects [20,21]. Medicinal herbs are an abundant source of diverse dietary polyphenols exhibiting prophylactic properties and therapeutic efficacy in various diseases including inflammatory- and obesity-associated disorders [19,22]. *Inula britannica* is a flowering wild plant distributed in eastern Asia and is used for medicinal purposes based on traditional applications. The flower of *I. britannica* exhibits neuroprotective, antimicrobial, anti-tumor, and anti-inflammatory properties [19,23,24], which are potentially attributable to its bio-active compounds such as flavonoids, sesquiterpene lactones, and polysaccharides [25,26,27,28]. For example, polysaccharides derived from the flower of *I. britannica* demonstrated hypoglycemic and anti-hyperlipidemic properties in an alloxan-induced diabetic mouse model [29]. However, the effects of *I. britannica* on adipogenesis and the underlying molecular mechanism have not been examined. Therefore, this study aimed to determine the anti-adipogenic effect of aqueous extract of *I. britannica* flower and its underlying mechanisms in the 3T3-L1 preadipocytes and to identify the phenolic compounds in the extract.

## 2. Materials and Methods

### 2.1. Chemicals and Reagents

Dulbecco’s modified Eagle’s medium (DMEM), water, trypsin-EDTA solution, phosphate-buffered saline (PBS), and antibiotics solution (penicillin, 10,000 U/mL; streptomycin, 10,000 μg/mL) were purchased from Hyclone Laboratories, Inc. (South Logan, UT, USA). Newborn calf serum (NCS), fetal bovine serum (FBS), and 0.4% trypan blue solution were purchased from Life Technologies (Carlsbad, CA, USA). Halt^TM^ protease and phosphatase inhibitor cocktails and reagents for quantitative real-time polymerase chain reaction (qRT-PCR) were purchased from Thermo Scientific Pierce (Waltham, MA, USA). Equipment and reagents for western blotting analysis were purchased from Bio-Rad (Hercules, CA, USA). The anti-FAS, anti-perilipin, anti-SCD-1, anti-FABP4, anti-adiponectin, anti-PPAR-γ, anti-C/EBP-α, anti-cyclin-dependent kinase (CDK)-4, anti-p27^KIP1^, anti-p-mitogen-activated protein kinase kinase (MEK) (Ser217/221), anti-ERK, anti-p-ERK (Thr202/Tyr204), anti-Akt, anti-p-Akt (Ser473), anti-mammalian target of rapamycin (mTOR), anti-p-mTOR (Ser2448), anti-p70S6K, anti-p-p70S6K (Ser371), anti-p-glycogen synthase kinase (GSK)-3β (Ser21/9), anti-p-cdc2 (Tyr15), and anti-p-STAT3 (Tyr705) primary antibodies and horseradish peroxidase (HRP)-conjugated secondary antibodies were purchased from Cell Signaling Technology, Inc. (Beverly, MA, USA). The other primary antibodies were purchased from Santa Cruz Biotechnology (Santa Cruz, CA, USA). Other chemicals including 3-isobutyl-1-methylxanthine (IBMX), dexamethasone (DEX), and insulin solution were purchased from Sigma-Aldrich (St. Louis, MO, USA).

### 2.2. Sample Preparation

The *I. britannica* flower aqueous extract (IAE) was prepared as previously described with minor modifications [24]. Briefly, commercially purchased dried flowers of *I. britannica* (Herb Kingdom Agriculture, Namwon, Korea) were pulverized and subjected to aqueous extraction (1:10; *I. britannica* powder:distilled water; w/v) for 72 h at 60 °C. The extract was filtered through a 0.45-μm filter, lyophilized, and stored at −20 °C until further use. The IAE was filtered through a 0.2-μm filter before use in subsequent experiments.

### 2.3. Cell Culture and Adipocyte Differentiation

The 3T3-L1 preadipocytes (American Type Culture Collection, Rockville, MD, USA) were cultured and maintained in growth medium (DMEM supplemented with 10% NCS and 1% antibiotic solution) at 5% CO_2_ and 37 °C in a humidified atmosphere. The differentiation of preadipocytes was performed as described previously [7]. Briefly, two-day-post-confluent 3T3-L1 preadipocytes (designated as day 0) were induced to differentiate by replacing the growth medium with the differentiation medium (MDI) comprising DMEM supplemented with 10% FBS, 1% antibiotics solution, adipogenic hormonal cocktail (0.5 mM IBMX, 1 μM DEX, and 5 μg/mL insulin). The cells were incubated for 48 h (day 0–2) and the culture medium was replenished with DMEM containing 10% FBS, 1% antibiotics solution, and 5 μg/mL insulin once every two days until day 8. During differentiation, the cells were treated with different concentrations (0–200 μg/mL) of IAE for periods indicated in the figure captions. To examine the mechanism underlying the anti-adipogenic effect of IAE, the cells were treated with 10 μM of specific inhibitors (U0126, ERK 1/2 inhibitor; LY294002, Akt inhibitor) for 1 h prior to MDI treatment. The undifferentiated and differentiated cells were defined as negative control groups and positive control groups, respectively.

### 2.4. Oil Red O Staining and Intracellular Triglyceride Quantification

The 3T3-L1 preadipocytes were plated in 60 mm cell culture dishes (1.0 × 10^5^ cells/dish) and differentiated into mature adipocytes as described in Section 2.3. The cells were rinsed twice with PBS and fixed with 4% neutral paraformaldehyde for 1 h at 4 °C. After washing with PBS, cells were maintained in 60% isopropyl alcohol for 5 min. The intracellular lipid droplets were stained with filtered (0.45 μm) 0.3% Oil Red O solution (w/v, in 60% isopropyl alcohol) for 1 h. Following the aspiration of residual Oil Red O solution, cells were rinsed five times with PBS. The images were acquired using a Nikon Eclipse E400-microscope/camera (Nikon, Tokyo, Japan). The Oil Red O dye was eluted with 100% isopropyl alcohol, and the absorbance was measured at 500 nm. The concentrations of intracellular triglycerides were assessed using a commercial kit (BioVision, Milpitas, CA, USA).

### 2.5. Cell Viability

Cell viability was assessed with 3-(4,5-dimethylthiazol-2-yl)-2,5-diphenyltetrazolium bromide (MTT) assay [30]. The cells plated in 24-well culture plates (2.0 × 10^4^ cells/well) were cultured in the growth medium for 48 h. The culture medium was replenished with the growth medium containing different concentrations of IAE and incubated for 48 h. The cells were then treated with MTT solution (dissolved in growth medium) at a final concentration of 0.5 mg/mL. Following the removal of the supernatant, the formazan deposits were dissolved in 1 mL of dimethyl sulfoxide. The absorbance was measured at 570 nm. Cell viability was represented as a percentage relative to that in the control groups. The effect of IAE on the cell viability of mature adipocytes was examined on day 8.

### 2.6. Immunoblotting

Western blotting analysis was performed as described previously with a minor modification [31]. The 3T3-L1 preadipocytes plated in 60 mm culture dishes (1.0 × 10^5^ cells/dish) were incubated and differentiated for various intervals in the presence of different concentrations of IAE. The cells were rinsed thrice with ice-cold PBS and lysed using a Pro-prep protein extraction buffer (iNtRON Biotechnology, Gyeonggi-do, Korea) supplemented with protease and phosphatase inhibitor cocktails. The lysates were sonicated (1 Amp; pulse-on, 5 s; pulse-off, 5 s) for 25 s in an ice-bath and centrifuged (15,000× *g*, 30 min, 4 °C). The protein concentration in the supernatant was determined using the DC^TM^ Protein Assay Kit (Bio-Rad). Equal amounts (25–40 μg) of cellular proteins were subjected to sodium dodecyl sulfate-polyacrylamide gel electrophoresis. The resolved proteins were transferred onto a polyvinyl difluoride membrane. The membrane was blocked with 5% skim milk and probed with the primary antibodies. Furthermore, the membrane was incubated with the HRP-conjugated secondary antibodies. The protein bands were detected using x-ray blue films and the enhanced chemiluminescence detection kit (Bio-Rad). The density of each protein band was measured using the ImageJ software (National Institutes of Health, Bethesda, MD, USA).

### 2.7. qRT-PCR

The qRT-PCR analysis was performed as described previously [31]. The 3T3-L1 preadipocytes seeded in 6-well culture plates (5 × 10^4^ cells/well) were differentiated until day 8, as described in Section 2.3. The cells were rinsed thrice with ice-cold PBS and subjected to RNA extraction using the commercial RNeasy Kit (Qiagen, Hilden, Germany). An equal amount (1 μg) of total RNA was reverse-transcribed to complementary DNA using a RevertAid^TM^ first-strand cDNA synthesis kit (Thermo Scientific Pierce). The qRT-PCR analysis was performed using SYBR Green PCR Master Mix (Thermo Scientific Pierce) in PikoReal 96 (Thermo Scientific Pierce), following the manufacturer’s instructions. The expression levels of target genes were normalized with those of the TATA box binding protein (TBP) [32]. The relative RNA expression levels were analyzed using the 2^−(ave.ΔΔCT)^ method. The melting curve of each gene was examined to verify the amplification of a single product. The primers used for qRT-PCR analysis are presented in Appendix A.

### 2.8. Trypan Blue Assay

Cell proliferation was evaluated with the trypan blue assay [11]. The 3T3-L1 preadipocytes plated in 6-well plates (5 × 10^4^ cells/well) were differentiated as described in Section 2.3. The cells were harvested after 0, 24, and 48 h of stimulation with MDI by using trypsin-EDTA. Following the centrifugation (500× *g*, 5 min, 4 °C), collected pellets were suspended in growth medium and stained with 0.4% trypan blue solution. The viable cell numbers were counted using a hemocytometer.

### 2.9. Fluorescence-Activated Cell Sorting (FACS) Analysis

Cell cycle progression during MCE was investigated using FACS [7]. The 3T3-L1 preadipocytes plated in 60 mm culture dishes (1.0 × 10^5^ cells/dish) were induced to differentiate, as described in Section 2.4. After stimulation with MDI for 16 h, cells were trypsinized and centrifuged (500× *g*, 5 min, 4 °C). The pellets were rinsed twice with PBS and fixed overnight in 70% ethyl alcohol at −20 °C. Following the removal of residual ethyl alcohol, cells were washed thrice with PBS and incubated in PBS containing 10 μg/mL RNase A and 50 μg/mL propidium iodide (PI) for 30 min. FACS analysis was performed using a CytoFLEX flow cytometer (Beckman Coulter, Brea, CA, USA). In total, 10,000 cells in each sample were used to measure the fluorescence of PI. The data were collected and analyzed by using CytExpert software (Beckman Coulter).

### 2.10. Ultra Performance Liquid Chromatography-Electrospray Ionization-Q/Orbitrap (UPLC-ESI-Q/Orbitrap) Tandem Mass Spectrometry (TEM), and High-Performance Liquid Chromatography (HPLC) Analysis

The UPLC-ESI-Q/Orbitrap tandem mass spectrometry analysis was conducted as previously described to identify the phenolic compounds of IAE [25]. Briefly, the UPLC analysis was performed using the Ultimate 3000 UPLC system (Thermo Fisher Scientific) equipped with a Hypersil GOLD^TM^ C18 column (2.1 mm × 100 mm, 1.9 μm; Thermo Fisher Scientific). The binary mobile phase system (A, 0.1% formic acid in water; B, 0.1% formic acid in acetonitrile) was employed with the linear gradient program at a flow rate of 0.2 mL/min. The injection volume was 1 μL. Following ionization in negative mode, five ions were analyzed by using a Q-Exactive Orbitrap mass spectrometer (Thermo Fisher Scientific). The data were analyzed using Xcalibur^TM^ software (Thermo Fisher Scientific).

Based on the results of UPLC-ESI-Q/Orbitrap tandem mass spectrometry analysis, the phenolic compounds of IAE were quantified using HPLC analysis [24]. Briefly, HPLC analysis was performed with Waters 600 HPLC system (Waters Corporation, Milford, MA, USA) equipped with an Eclipse XDB-C18 column (4.6 mm × 150 mm, 5 μm; Agilent Technologies, Santa Clara, MA, USA) and a Waters 2487 Dual-wavelength detector (Waters Corporation). The concentration of each compound was determined from the external regression curve, which was constructed with five concentrations of standards.

### 2.11. Statistical Analysis

The data were presented as mean ± standard deviation of three values obtained from the independent experiment conducted in triplicate at least. The values of respective experiments were determined by the mean of three measurements. The data were analyzed by using IBM SPSS version 24.0 (SPSS Inc., Chicago, IL, USA). Differences among multiple groups were evaluated with a one-tailed one-way analysis of variance (one-way ANOVA), followed by Tukey’s multiple comparison test (*p* < 0.05). The differences were considered significant at *p* < 0.05.

## 3. Results

### 3.1. IAE Inhibits the Lipid Accumulation without Inducing Cytotoxic Effects

The dose-escalating cytotoxicity assessment was performed using the MTT assay. Compared with that in the control groups, IAE did not show significant adverse effect on the cell viability of undifferentiated preadipocytes or differentiated adipocytes up to concentrations of 500 μg/mL (Figure 1A). The effect of IAE on lipid accumulation in the differentiated adipocytes was examined. The Oil Red O staining showed that treatment with IAE dose-dependently inhibited the MDI-induced intracellular lipid accumulation (Figure 1B,C). Correspondingly, the intracellular triglyceride levels were significantly decreased upon treatment with IAE (Figure 1D). These results indicated that IAE inhibits lipogenesis in the MDI-induced 3T3-L1 preadipocytes without exerting cytotoxic effects. Therefore, the cytotoxicity of IAE was not considered in the subsequent experiments.

### 3.2. IAE Exhibits Anti-Adipogenic Effect during Adipocyte Differentiation

To evaluate the effect of IAE on adipogenesis, the expression levels of lipogenesis- and adipogenesis-associated mediators were investigated in differentiated 3T3-L1 cells upon treatment with IAE. First, the effect of IAE on lipogenesis-related biomarkers was assessed. Consistent with the results of intracellular lipid contents, treatment with MDI markedly increased the protein expression levels of FAS, FABP4, perilipin, and SCD-1. However, the expression levels of those proteins were mitigated upon treatment with IAE (Figure 2A,B). Consistently, treatment with IAE downregulated the mRNA expression levels of FAS, FABP4, perilipin, and SCD-1 upon MDI stimulation (Appendix A). Following this, the effect of IAE on the expression of adipogenesis-specific biomarkers was examined. As expected, IAE treatment significantly suppressed the expression levels of PPAR-γ, C/EBP-α, sterol regulatory element-binding protein (SREBP)-1c, and adiponectin following MDI treatment at both the protein (Figure 2C,D) and transcriptional levels (Appendix A). These findings demonstrated that IAE inhibits the de novo lipogenesis by regulating the adipogenesis, which results in the inhibition of the accumulation of intracellular lipids.

To further examine the anti-adipogenic properties of IAE, the effect of IAE on the activation of the Akt/GSK-3β axis was investigated in the 3T3-L1 cells on day 8. Upon stimulation with insulin, the Akt/GSK-3β signaling pathways promote the terminal differentiation of adipocytes through the phosphorylation of C/EBP-α and modulation of glucose uptake [33]. Consistent with the preceding results, the MDI-induced phosphorylation of Akt and GSK-3β was significantly attenuated upon treatment with IAE (Figure 2E,F). These results indicated that IAE inhibits adipogenesis and subsequent intracellular lipid accumulation in differentiated 3T3-L1 cells through regulating the Akt/GSK-3β signaling pathways.

### 3.3. IAE Modulates the Early Phase of Adipogenesis

To elucidate the mechanism underlying the anti-adipogenic effect of IAE, the 3T3-L1 cells were treated with IAE at different time points during differentiation (Figure 3A). Compared with the control groups (treatment no. 1), treatment with IAE reduced the intracellular lipid accumulation irrespective of the treatment periods (Figure 3B,C). Additionally, the strongest inhibition of lipid accumulation was observed upon treatment with IAE during the first two days. The lipid accumulation in the cells treated with IAE for the first two days (treatment no. 3) was comparable to that in the cells treated with IAE for eight days (treatment no. 2). Consistent with the result of Oil Red O staining, the protein expression levels of FAS, PPAR-γ, C/EBP-α, and p-Akt were significantly downregulated upon treatment with IAE in MDI-stimulated 3T3-L1 preadipocytes (Figure 3D), which indicated that IAE modulates the early phase of adipogenesis. Based on these findings, we hypothesized that the anti-adipogenic effect of IAE was closely associated with the regulation of MCE, which is critical for the initiation of adipogenesis.

### 3.4. IAE Inhibits the MCE by Arresting the Cell Cycle Progression

During the early phase of adipogenesis, MDI induction led the growth-arrested 3T3-L1 preadipocytes to undergo MCE, which involves synchronous re-entry into the cell cycle and cell proliferation [7]. Therefore, the effect of IAE on MDI-induced cell proliferation and cell cycle progression was investigated during early phases of 3T3-L1 preadipocyte differentiation. The result of the trypan blue assay revealed that treatment of IAE significantly inhibited the MDI-induced cell proliferation showing a 24.35% and 32.60% inhibition rate at 24 h and 48 h, respectively (Appendix A). Compared with the negative control group, the MDI-induced group exhibited 1.8- and 2.2-times higher cell numbers after 24 and 48 h, respectively, while the IAE (200 μg/mL)-treated group exhibited 1.4- and 1.5-times higher cell numbers at 24 and 48 h, respectively.

The role of cell cycle arrest in the inhibitory effect of IAE on cell proliferation was examined using FACS analysis and western blotting. The cell cycle of fully confluent 3T3-L1 preadipocytes was primarily remained at the G0/G1 phase. However, MDI stimulation promoted the entry of 3T3-L1 preadipocytes into the S phase. The analysis of cellular DNA content revealed that the proportions of cells at the G0/G1 and S phases in the negative control group were 78.05% and 6.98%, respectively, while those in the positive group were 51.56%, and 22.87%, respectively. This indicated that MDI induced a cell cycle transition from the G0/G1 phase to the S phase. However, IAE dose-dependently inhibited shift of cell cycle from the G0/G1 phase to the S phase (Figure 4A,B), as evidenced by a significant increase in the cell population remaining at the G0/G1 phase (ranging from 58.25 to 77.94%). In particular, the proportion of cells at S phase in the IAE (200 μg/mL)-treated group was 5.38%, which was comparable with that in the negative control groups. Next, the expression levels of cell cycle progression-associated proteins were evaluated. Consistent with the results of the FACS analysis, treatment with IAE suppressed the expression levels of cell cycle progression-related proteins in MDI-stimulated 3T3-L1 cells (Figure 4C,D). In particular, IAE dose-dependently suppressed the expression levels of S phase-mediating proteins (cyclin D1, cyclin A1, and CDK2) and significantly mitigated the degradation of p27 (CDK inhibitor 1B). Additionally, downregulated expression levels of G2/M-mediating proteins (cyclin B1 and p-cdc2) upon treatment of IAE corroborate the evidence that IAE inhibits cell cycle progression by arresting the cell cycle at the G0/G1 phase.

To further examine the cellular mechanisms involved in cell cycle progression, the effect of IAE on MDI-induced activation of mTOR/Akt/p70S6K axis was investigated. The mTOR/Akt/p70S6K axis plays a pivotal role in cell proliferation and metabolism by regulating the cell cycle progression and lipogenesis [34]. Consistent with previous findings, MDI stimulation markedly up-regulated the phosphorylation of mTOR/Akt/p70S6K (Figure 4E,F). However, IAE dose-dependently alleviated the MDI-induced phosphorylation of mTOR/Akt/p70S6K, which indicated that IAE inhibits MCE by causing the cell cycle arrest at the G0/G1 phase involving the mTOR/Akt/p70S6K signaling pathways. Based on these results, we hypothesized that the anti-mitotic effect of IAE is mediated through regulation of transcription factors involved in MCE during the early phases of adipogenesis.

### 3.5. IAE Regulates the Transcription Factors of the Early Phase of Adipogenesis

To determine whether the anti-mitotic effect was associated with alteration in the levels of adipogenic transcription factors, the effect of IAE on the activation of STAT3 and the expression of C/EBP-β was investigated in the MDI-stimulated 3T3-L1 preadipocytes during the early phase of differentiation. The activation of STAT3 and C/EBP-β mediates cell proliferation and activation of adipogenic transcription factors, thereby promoting MCE and adipogenesis [17,18,35]. Consistent with these findings, stimulation with MDI induced the phosphorylation of STAT3 and IAE attenuated the MDI-induced phosphorylation of STAT3 (Figure 5A). Additionally, IAE dose-dependently mitigated the MDI-induced phosphorylation of STAT3 (Figure 5B,C), which concurred with arrested cell cycle progression (Figure 4A–C) and downregulated expression of adipogenic transcription factors (Figure 2C and Appendix A) following MDI stimulation in the presence of IAE. Similarly, IAE attenuated the MDI-induced expression of C/EBP-β isoforms, C/EBP-β/LAP, and C/EBP-β/LIP (Figure 5D–F). Both C/EBP-β isoforms were rapidly expressed following MDI stimulation and treatment with IAE mitigated the MDI-induced expression of C/EBP-β isoforms (Figure 5D). In addition, IAE showed dose-dependent inhibitory effects on the MDI-induced C/EBP-β expression (Figure 5E,F). These findings indicated that the inhibitory effect of IAE on adipogenesis of 3T3-L1 preadipocytes is associated with regulation of STAT3 and C/EBP-β.

### 3.6. IAE Inhibits the Adipogenesis by Regulating the ERK 1/2 and Akt Signaling Pathways

The proliferation and differentiation of cells are modulated by cascades of transcription factors involving various cellular signaling pathways [9]. Insulin-like growth factor (IGF)-1, which is upregulated upon stimulation with MDI, activates the MEK-1/ERK 1/2 and Akt signaling pathways and consequently promotes activation of the adipogenic transcription factors [13,15,16]. Thus, the effect of IAE on MDI-induced activation of ERK 1/2 and Akt was investigated during the early phase of adipogenesis. Compared to MDI-treated groups, MDI-induced activation of MEK-1, ERK 1/2, and Akt was attenuated upon treatment with IAE during early phases of differentiation (Figure 6A). Additionally, IAE exhibited a dose-dependent inhibitory effects on the phosphorylation of MEK-1, ERK 1/2, and Akt (Figure 6B,C). Next, the ERK 1/2 and Akt signaling pathways in adipogenesis of 3T3-L1 preadipocytes confirmed by using U0126 and LY294002 inhibitors, respectively. The U0126 inhibits the MDI-induced ERK 1/2 activation by inactivating MEK-1 and MEK-2, which are the kinase of ERK 1/2, and consequently suppresses the activation of PPAR-γ and C/EBP-α [15]. Similarly, inactivation of Akt upon treatment LY294002 results in the inhibition of adipogenesis as well as lipogenesis [16]. Consistent with previous findings, the inhibition of ERK 1/2 and Akt upon pre-treatment with U0126 and LY294002 (Appendix A) resulted in suppression of MCE by arresting the cell cycle progression (Appendix A and Figure 6D) and mitigating the STAT3 activation (Figure 6E). Following this, U0126 and LY294002 inhibited the accumulation of intracellular lipids (Appendix A) and expression of adipogenic biomarkers, FAS, PPAR-γ, and C/EBP-α (Figure 6F). These results suggest that the anti-adipogenic effect of IAE is associated with regulation of the MEK-1/ERK 1/2 and Akt signaling pathways.

### 3.7. Determination of Phenolic Compounds of IAE

The phenolic compounds of IAE were examined using UPLC-ESI-Q/Orbitrap tandem mass spectrometry analysis and quantified using HPLC analysis (Table 1). IAE comprised gallic acid (m/z [M-H] = 169.08749), protocatechuic acid (m/z [M-H] = 153.00632), chlorogenic acid (m/z [M-H] = 353.34218), kaempferol-3-*O*-glucoside (m/z [M-H] = 447.35581), and 6-methoxyluteolin (m/z [M-H] = 315.25656), which was consistent with the results of previous studies on the major phenolic compounds of *I. britannica* flower [24,25,26,27,28]. Additionally, these phenolic compounds were reported to exhibit anti-obesogenic properties through the regulation of adipogenesis and lipogenesis [22]. The phenolic compounds were quantified using a HPLC system. The most abundant phenolic compound was kaempferol-3-*O*-glucoside (54.842 ± 0.191 μg/mg), followed by chlorogenic acid (16.540 ± 0.094 μg/mg), 6-methoxyluteolin (6.669 ± 0.286 μg/mg), and protocatechuic acid (3.998 ± 0.027 μg/mg). However, the levels of gallic acid were lower than the limit of quantification. These results suggest that IAE suppresses lipid accumulation by inhibiting the development of adipogenesis, which may be mediated by identified anti-obesogenic phenolic compounds in the extract.

## 4. Discussion

Obesity is reported to be associated with increased risk of onset of various degenerative diseases including metabolic syndrome morbidities. The aberrant growth of adipose tissue, which is a characteristic feature of obesity, is dependent on the hypertrophy and/or hyperplasia of adipocytes [7]. Adipocyte hypertrophy is an enlargement in size of pre-existed mature adipocytes, resulting from an accumulation of intracellular lipids in individual adipocytes. In contrast, adipocyte hyperplasia is an increased number of adipocytes due to the development of new adipocytes, which results from adipogenesis [36]. Adipocyte hyperplasia is a crucial process in determining the adipocyte numbers, which is primarily established in childhood and adolescence. The adipocyte number is sustained in adults even after weight loss, which indicates weight loss mainly results from the decreased volume of established adipocytes [22]. Following this, upregulated adipogenesis leads to the growth of adipose tissue by promoting the hyperplasia and hypertrophy of adipocytes and consequently results in the development of obesity. Therefore, the regulation of adipogenesis is a potential therapeutic strategy for obesity. Plant materials, which are a source of phenolic compounds with anti-obesogenic efficacy, are of increasing interest for prophylactic and therapeutic options in obesity and obesity-related disorders [37,38,39]. This study revealed the anti-obesity potential of *I. britannica* by demonstrating the anti-adipogenic effect of IAE and its underlying cellular mechanisms in the 3T3-L1 preadipocytes. Additionally, phenolic constituents of IAE were identified.

The differentiation of adipocytes accompanies upregulated de novo lipogenesis, which is modulated by various transcription factors such as C/EBPs and PPARs and results in the accumulation of intracellular lipids [40]. In particular, C/EBP-α and PPAR-γ are both anti-mitotic and essential for the development of adipogenesis, in addition to the maintenance of a differentiated state. These transcription factors terminate the MCE and reciprocally promote triacylglycerol synthesis, which is dependent on the transcriptional modulation of lipogenesis-associated genes such as *Srebp1c* (SREBP-1c), *Fasn* (FAS), *Fabp4* (FABP4), *Plin1* (perilipin), and *Scd1* (SCD-1), thereby establishing an adipocyte phenotype [12,22]. Numerous studies have demonstrated that these transcription factors play critical roles in adipogenesis and the development of obesity. For example, a previous study reported that the white adipose tissues of C/EBP-α knockout mice exhibited deficient metabolism and lipid storage [41]. Similarly, mice with adipocyte-specific deletion of PPAR-γ were resistant to high-fat diet (HFD)-induced obesity as the development of adipose tissue was impaired [42]. Therefore, the inhibition of these transcription factors is a potential therapeutic strategy for obesity with respect to the regulation of adipogenesis. Consistently, the *Edgeworthia gardineri* flower, *Aster spathulifolius*, and *Viburnum opulus* fruit extracts have been reported to inhibit development of adipogenesis by regulating the expression of C/EBP-α and PPAR-γ, which results in decreased accumulation of intracellular lipids [37,40,43]. Our data demonstrated that IAE inhibited intracellular lipid accumulation through suppressing the expression levels of lipogenesis- and adipogenesis-associated biomarkers including C/EBP-α and PPAR-γ. Additionally, the IAE-mediated inhibition of lipid accumulation was associated with the regulation of early phases of adipogenesis. This indicated that IAE inhibits intracellular lipid accumulation by suppressing adipogenesis rather than direct regulation of lipogenesis.

During the early phases of adipogenesis, quiescent preadipocytes are induced to undergo MCE upon hormonal stimulation. Hence, growth-arrested preadipocytes synchronously re-enter the cell cycle and consequently undergo one or two rounds of mitosis, which increases the numbers of adipocytes and results in adipocyte hyperplasia [9,22]. Despite several controversies regarding the role of MCE in adipogenesis, numerous studies have reported that MCE is a prerequisite for the terminal phases of adipogenesis by demonstrating the correlation between MCE and adipogenesis, which is dependent on the activation of adipogenic transcriptional cascades [13,18,44,45]. C/EBP-β and STAT3, which are early adipogenic transcription factors, are involved in development of MCE and adipogenesis [17,46]. Activated C/EBP-β develops MCE by mediating DNA replication and promotes adipogenesis by activating the transcription of C/EBP-α and PPAR-γ. Consistently, 3T3-L1 preadipocytes following knockdown of C/EBP-β genes, neither undergo MCE nor develop into adipocytes [46,47]. In addition to activating C/EBP-β and PPAR-γ, STAT3 is involved in cell proliferation by promoting cell cycle progression [7,17]. The growth-arrested preadipocytes remained predominantly at the G0/G1 phase due to upregulated cell cycle suppressor proteins such as hypo-phosphorylated retinoblastoma (Rb), a tumor suppressor protein, and p27^KIP1^, a CDK inhibitor protein. However, hormonal stimulation induces the degradation of CDK inhibitor proteins and consequently promotes cell cycle progression by sequential activation and assembly of cyclin D1/CDK4/CDK6, cyclin E/CDK2, cyclin A1/CDK2, and cyclin B1/CDK1 (cdc2) complexes. In particular, cyclin E/CDK2 and cyclin A1/CDK2 complexes play crucial roles in cell cycle progression by mediating the G1/S phase transition and S phase progression. Furthermore, C/EBP-β and STAT3 are activated by acquiring DNA-binding capacity during the S phase [7,12,22,46]. A previous study has demonstrated that cell cycle arrest leads to the disruption of transactivation domains of C/EBP-β, which results in the inhibition of adipogenesis [18]. Several studies have demonstrated that the upregulated expression of p27^KIP1^ contributes to the inhibition of MCE and adipocyte differentiation, whereas the degradation of p27^KIP1^ promotes adipogenesis and increases fat mass, indicating that p27^KIP1^ potentially inhibits hyperplasia [48,49]. Therefore, inhibition or delaying the cell cycle progression by regulating the cell cycle modulators during MCE as well as inactivating the early transcription factors could be efficacious strategies to inhibit adipogenesis and mitigate obesity. Our data showed that IAE inhibited cell proliferation by arresting the cell cycle at the G0/G1 phase, which was corroborated by inactivated cell cycle modulators and upregulated expression levels of p27^KIP1^. Additionally, IAE inhibited the expression of C/EBP-β and the activation of STAT3. Consistent with preceding results, previous studies have reported that plant extracts and their phytochemicals inhibit adipogenesis and lipid accumulation through suppression of MCE, which was dependent on the regulation of cell cycle progression and/or suppression of transcription factors [50,51,52]. For instance, *Gleditsia sinensis* fruit extract suppressed MCE and adipogenesis through the inhibition of p27^KIP1^ degradation and the activation of STAT3 [7].

The ERK 1/2 and Akt cellular signaling pathways are involved in MCE and adipogenesis, depending on the modulation of proliferation, differentiation, and lipid metabolism of adipocytes [11,33,34,53]. The activation of the MEK-1/ERK 1/2 signaling pathways promotes cell cycle progression by activating the cell cycle modulators such as the cyclin D1 and Rb [15,53]. Similarly, the activation of the mTOR/Akt/p70S6K axis mediates cell proliferation by modulating cyclin D1 and p27^KIP1^ [34]. Additionally, the phosphorylation cascades of MEK-1/ERK 1/2 and Akt/GSK-3β sequentially lead to the phosphorylation and acquisition of DNA-binding activity of C/EBP-β, which promotes adipogenesis [18,22,33]. Thus, the ERK 1/2 and Akt signaling pathways are potential molecular therapeutic targets for obesity. Correspondingly, mice with the deletion of ERK genes are resistant to HFD-induced obesity, which was consistent with the suppressed expression of C/EBP-α and PPAR-γ in 3T3-L1 preadipocytes upon blockage of MEK/ERK cascades [15,49]. Our data indicated that IAE inhibits adipogenesis including MCE by regulating the activation of the ERK 1/2 and Akt signaling pathways. Consistent with these results, *Clitoria ternatea* flower and coffee extracts inhibited MDI-induced phosphorylation of ERK 1/2 and Akt, which resulted in the inhibition of adipogenesis development [41,50].

Phenolic compounds including phenolic acids and flavonoids have been demonstrated to exhibit anti-obesogenic properties such as the suppression of adipogenesis and the degradation of intracellular lipids [22,38,39]. The phytochemical analysis of IAE revealed the presence of protocatechuic acid, chlorogenic acid, kaempferol-3-*O*-glucoside, and 6-methoxyluteolin. Dietary flavonoids such as kaempferol, quercetin, and curcumins inhibited the adipogenesis of 3T3-L1 preadipocytes by suppressing MCE and regulating adipogenic transcriptional cascades [22,49]. For instance, kaempferol induced cell cycle arrest at the S phase by inactivating the mTOR/p70S6K/Akt axis, thereby inhibiting the adipogenesis accompanied by the downregulated expression of C/EBP-α and PPAR-γ [34]. Similarly, phenolic acids such as chlorogenic acid, ellagic acid, and *p*-coumaric acid suppressed intracellular lipid accumulation by inhibiting the expression of lipogenic enzymes, which results from the downregulated expression of C/EBP-α and PPAR-γ [18,39]. Moreover, previous studies have demonstrated that chlorogenic acid and luteolin alleviate obesity in HFD-induced mice by improving lipid metabolism, in addition to exercise endurance [54,55]. Additionally, flavonoids and phenolic acids regulate the activity of PPAR-γ by interacting with receptors as an agonist, which is attributable to their amino acid residue-dependent binding affinity to PPAR-γ, thereby inhibiting adipogenesis [39]. Our results indicate that anti-obesogenic phenolic compounds are potentially involved in mediating the anti-adipogenic effect of IAE.

## 5. Conclusions

This study revealed the anti-obesity potential of IAE by demonstrating the anti-adipogenic effects of IAE and elucidating the underlying mechanisms during the differentiation of 3T3-L1 preadipocytes. The inhibitory effect of IAE on MCE during the early phases of adipogenesis primarily contributed to the suppressed expression of adipogenesis-associated biomarkers, which resulted in the inhibition of adipogenesis and lipogenesis. Additionally, IAE exerted an anti-adipogenic effect through the regulation of the ERK 1/2 and Akt cellular signaling pathways. Furthermore, IAE comprised anti-obesogenic phenolic compounds. Although further in vivo studies are needed to confirm the anti-obesity properties of IAE, the findings of this study indicate that IAE can potentially prevent obesity by regulating the hyperplasia and hypertrophy of adipocytes. Thus, IAE might be considered as a potential therapeutic agent for obesity. Overall, IAE could be potentially used for the development of prophylactic and therapeutic products for obesity.

## Figures and Tables

**Figure 1 nutrients-12-03037-f001:**
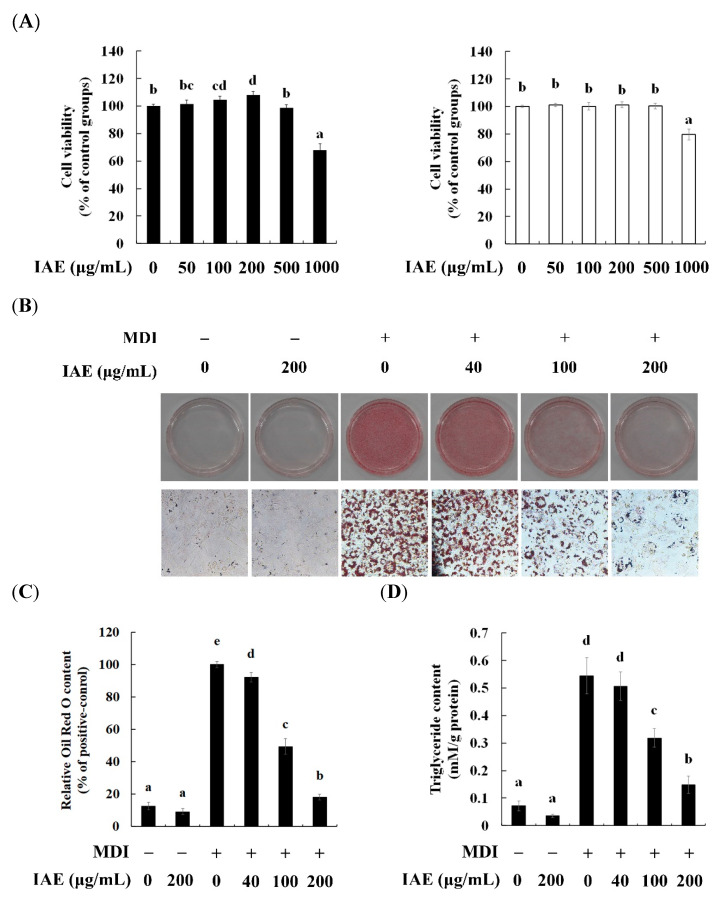
*Inula britannica* flower aqueous extract (IAE) inhibits lipid accumulation without exerting cytotoxic effects during the differentiation of 3T3-L1 preadipocytes. The cells were induced to differentiate by the MDI (differentiation medium) upon treatment with IAE in indicated concentrations. (**A**) The effect of IAE on cell viability was evaluated with the MTT assay (■, cell viability of preadipocytes; □, cell viability of differentiated preadipocytes). (**B**) Oil Red O staining of intracellular lipids on day 8. (**C**) Relative absorbance of Oil Red O eluted from intracellular lipids at 500 nm. (**D**) Measurement of intracellular triglyceride levels on day 8. The data are presented as mean ± standard deviation. Values labeled with different letters (a–e) are significantly different (*p* < 0.05). Differences among the multiple groups were determined based on one-tailed one-way analysis of variance, followed by Tukey’s post hoc test.

**Figure 2 nutrients-12-03037-f002:**
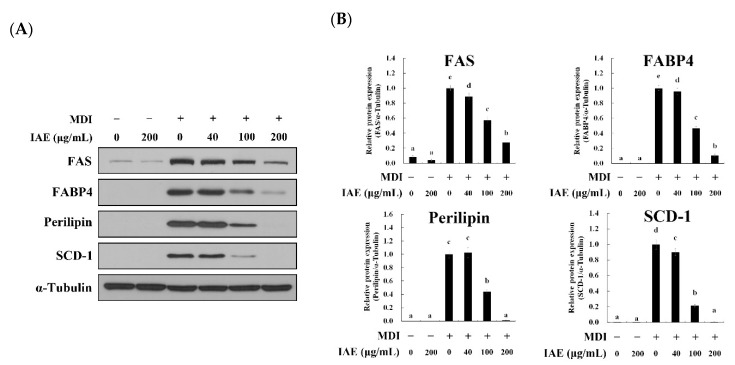
*Inula britannica* flower aqueous extract (IAE) exerts anti-adipogenic effects in MDI-induced differentiation of 3T3-L1 preadipocytes involving regulation of Akt/GSK-3β signaling pathways. The cells were differentiated for eight days with the indicated concentrations of IAE. The phosphorylation of Akt (Ser 473) and GSK-3β (Ser21/9) was examined to evaluate the activation of the Akt/GSK-3β signaling pathways. The protein expression levels of adipogenesis-associated biomarkers were determined by using western blotting. (**A**) Protein expression levels of lipogenesis-associated biomarkers. (**B**) Relative protein expression levels of lipogenesis-associated biomarkers. (**C**) Protein expression levels of adipogenesis-specific biomarkers. (**D**) Relative protein expression levels of adipogenesis-specific biomarkers. (**E**) Effect of IAE on the activation of Akt/GSK-3β signaling pathways. (**F**) Relative protein expression levels of phosphorylated Akt and GSK-3β. The data are presented as mean ± standard deviation. Values labeled with different letters (a–e) are significantly different (*p* < 0.05). Differences among the multiple groups were determined based on one-tailed one-way analysis of variance, followed by Tukey’s post hoc test.

**Figure 3 nutrients-12-03037-f003:**
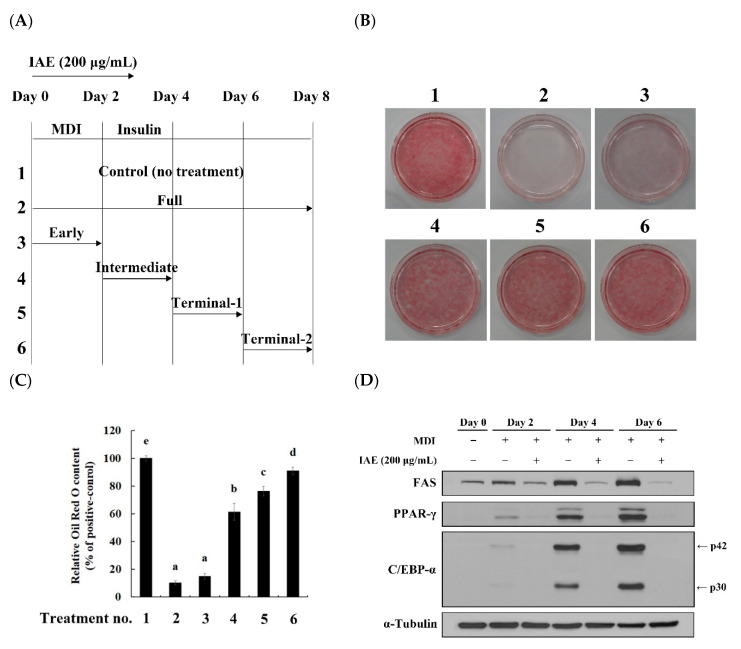
*Inula britannica* flower aqueous extract (IAE) affects the early phases of adipogenesis. During the eight days of 3T3-L1 differentiation, IAE (200 μg/mL) was treated at different time points as indicated. (**A**) Schematic depiction of different treatment intervals of IAE during 3T3-L1 differentiation. (**B**) Intracellular lipids were stained with Oil Red O on day 8. (**C**) Relative absorbance of Oil Red O dye eluted from intracellular lipids at 500 nm. (**D**) Expression levels of adipogenesis-associated proteins on day 0, 2, 4, and 6 of differentiation. The data are presented as mean ± standard deviation. Values labeled with different letters (a–d) are significantly different (*p* < 0.05). Differences among the multiple groups were determined based on one-tailed one-way analysis of variance, followed by Tukey’s post hoc test.

**Figure 4 nutrients-12-03037-f004:**
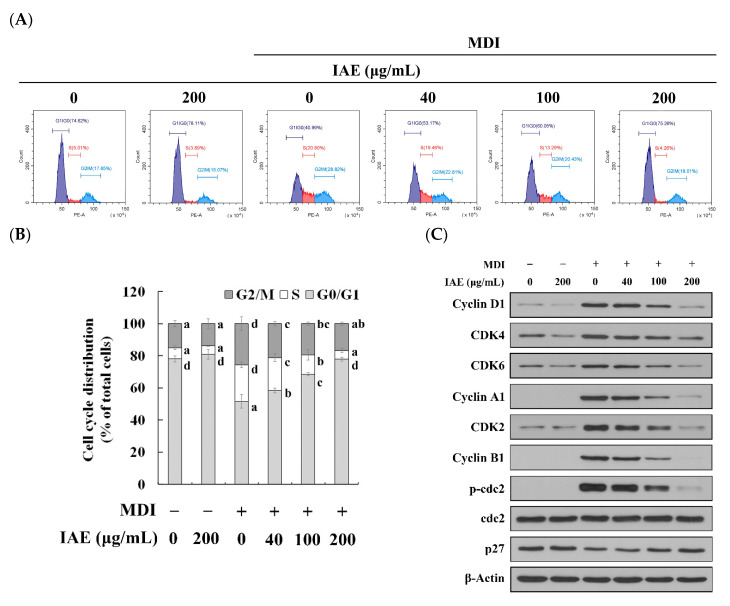
*Inula britannica* flower aqueous extract (IAE) inhibits the MCE involving Akt signaling pathways during early phase of adipogenesis. Growth-arrested 3T3-L1 preadipocytes were induced to differentiate by stimulating with MDI in the presence of indicated concentrations of IAE. The phosphorylation of cdc2 (Tyr15) and mTOR (Ser2448), Akt (Ser473), and p70S6K (Ser371) was examined to evaluate the activation of the G2/M phase and mTOR/Akt/p70S6K axis, respectively. (**A**) After 16 h of MDI treatment, cell cycle progression was investigated using FACS. (**B**) The cell distribution in the G0/G1, S, and G2/M phases was calculated as a percentage of total cell numbers based on the results of FACS analysis. (**C**) Western blotting was used to examine the expression of cell cycle progression-associated proteins after 18 h of MDI treatment. (**D**) Relative expression levels of representative proteins contributing to the transition of the cell cycle. (**E**) Effect of IAE on the activation of the mTOR/Akt/p70S6K axis after 18 h of MDI stimulation. (**F**) Relative expression levels of phosphorylated mTOR, Akt, and p70S6K. The data are presented as mean ± standard deviation. Values labeled with different letters (a–d) are significantly different (*p* < 0.05). Differences among the multiple groups were determined based on one-tailed one-way analysis of variance, followed by Tukey’s post hoc test.

**Figure 5 nutrients-12-03037-f005:**
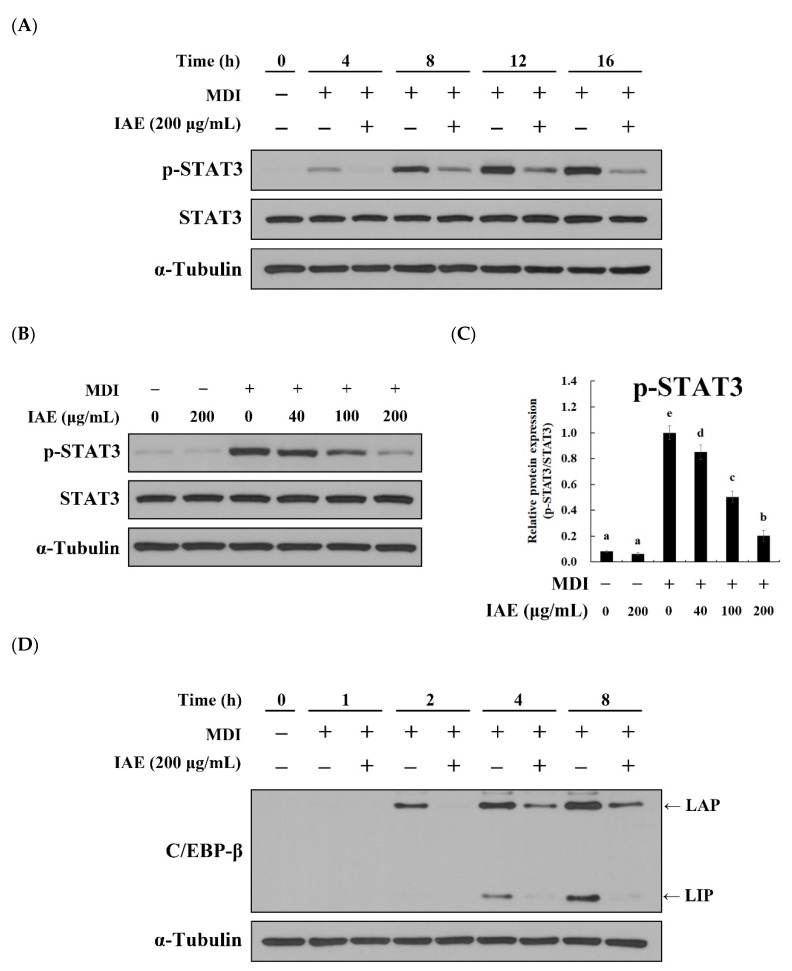
*Inula britannica* flower aqueous extract (IAE) inhibits the activation of STAT3 and C/EBP-β during the early phase of adipogenesis. Fully confluent 3T3-L1 preadipocytes were differentiated by MDI stimulation upon treatment with indicated concentrations of IAE. The phosphorylation of STAT3 (Tyr705) was examined to evaluate STAT3 activation. The expression levels of proteins were assessed with western blotting. (**A**) The time-course of MDI-induced STAT3 phosphorylation in the presence or absence of IAE. (**B**) The dose-dependent inhibitory effect of IAE on STAT3 activation was determined after 18 h of MDI stimulation. (**C**) Relative protein expressions of phosphorylated STAT3. (**D**) The time-course of MDI-induced C/EBP-β expression in the presence or absence of IAE. (**E**) The dose-dependent inhibitory effect of IAE on C/EBP-β was assessed after 4 h of MDI stimulation. (**F**) Relative protein expression levels of C/EBP-β. The data are presented as mean ± standard deviation. Values labeled with different letters (a–e) are significantly different (*p* < 0.05). Differences among the multiple groups were determined based on one-tailed one-way analysis of variance, followed by Tukey’s post hoc test.

**Figure 6 nutrients-12-03037-f006:**
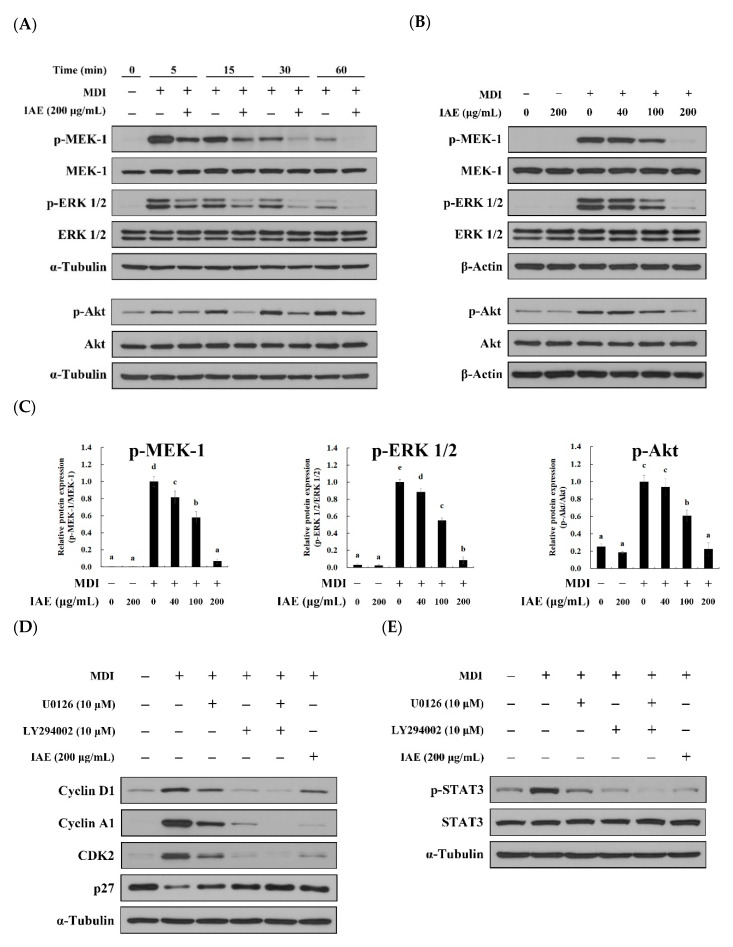
*Inula britannica* flower aqueous extract (IAE) inhibits the adipogenesis of 3T3-L1 preadipocytes through regulation of the ERK 1/2 and Akt signaling pathways. The 3T3-L1 preadipocytes were induced to differentiate by MDI in the presence of indicated concentrations of IAE or specific inhibitors (U0126 and LY294002). Cells were treated with specific inhibitors for 1 h prior to MDI stimulation. The phosphorylation of MEK-1 (Ser217/221), ERK 1/2 (Thr202/Tyr204), and Akt (Ser473) was examined to evaluate activation of these proteins. Western blotting was performed to examine the expression levels of proteins. (**A**) Inhibitory effect of IAE on activation of MEK-1/ERK and Akt during early phase of adipogenesis. (**B**) The dose-dependent inhibitory effect of IAE on MDI-induced MEK/Akt activation was investigated following 30 min of MDI induction. (**C**) Relative protein expression levels of MEK-1, ERK 1/2, and Akt. (**D**,**E**) After 18 h of MDI treatment, the effect of ERK 1/2 and Akt signaling pathways on the expression of proteins mediating the cell cycle progression was assessed. (**F**) Effect of ERK 1/2 and Akt signaling pathways on adipogenesis-associated biomarkers was assessed on day 8. The data are presented as mean ± standard deviation. Values labeled with different letters (a–e) are significantly different (*p* < 0.05). Differences among the multiple groups were determined based on one-tailed one-way analysis of variance, followed by Tukey’s post hoc test.

**Table 1 nutrients-12-03037-t001:** Ultra-performance liquid chromatography-electrospray ionization-Q/Orbitrap tandem mass spectrometry analysis of *Inula britannica* flower aqueous extract.

Compounds	Retention Time (min)	m/z [M-H]	MS^2^ Fragment	Molecular Formula	Contents (μg/mg)	References
Gallic acid	0.08	169.08749	125.09668	C_7_H_6_O_5_	<LOQ ^1^	
Protocatechuic acid	0.34	153.00632	109.06452	C_7_H_6_O_4_	3.998 ± 0.027	[25,26]
Chlorogenic acid	3.65	353.34218	168.98956	C_16_H_18_O_9_	16.540 ± 0.094	[25,27]
Kaempferol-3-*O*-glucoside	4.13	447.35581	168.98952	C_21_H_20_O_11_	54.842 ± 0.191	[25]
6-Methoxyluteolin	14.50	315.25656	96.96816	C_16_H_12_O_7_	6.669 ± 0.286	[25]

The experiment was independently performed in triplicate. The concentration of phenolic compounds is presented as mean ± standard deviation. ^1^ LOQ, limit of quantification.

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
