# Peer review of "Inula britannica Inhibits Adipogenesis of 3T3-L1 Preadipocytes via Modulation of Mitotic Clonal Expansion Involving ERK 1/2 and Akt Signaling Pathways"

_nutrients, 2020, doi:10.3390/nu12103037_

Round 1
Reviewer 1 Report
Comments for Authors:
The study conducted by Hyung-Seok et al. aims to investigate the anti-adipogenic properties of aqueous extract of I. britannica (IAE) on 3T3-L1 preadipocytes and its potential in preventing lipid accumulation. The methodology was extensive and proper, and the data generated point to an anti-obesogenic potential of IAE. It was shown that IAE could effectively downregulate the expression of biomarkers associated with adipogenesis, inhibit cell cycle progression, and regulate the ERK1/2 and Akt signaling pathways. Despite these data, certain points must be addressed as shown below, and extensive proof-reading must be conducted to improve the overall quality of the article.
Major Comments:
The manuscript requires extensive editing. English grammar and syntax are not followed.
Line 422-425 “Collectively, IAE suppresses lipid accumulation
by inhibiting the development of adipogenesis, which is attributable to its anti-obesogenic phenolic
Line 542: “these findings, our results indicated that the anti-adipogenesis effect of IAE was might be associated with anti-obesogenic phenolic compounds of IAE”
The above statements are not supported by any data. The authors just measured the content/levels of a few polyphenols in their extract. They never tested these polyphenols and therefore the text must be changed to reflect the data accurately.
Introduction: Include statistics/epidemiology of obesity, risk factors, and provide information regarding the role of adipose tissue in healthy individuals (few sentences).
Section 2.8 of the methodology: Must verify if results are accurate. How many independent experiments were carried out? How is error accounted for when performing an assay involving quantification following trypsinization? Cells may be lost after washing or depending on time of exposure to trypsin, favouring penetration of trypan blue.
Section 3.5. The authors state that IAE caused a time-dependent reduction in MDI-induced expression of p-STAT3 and C/EPB-β. The representative blots in Fig. 5C and 5F do not adequately reflect this. If IAE time-dependently inhibits the effect of MDI, this should be reflected as a more pronounced reduction in p-STAT3 and C/EPB-β at higher treatment times however this is not what the blots show. Graphs showing densitometry and statistical significance must be added.
Section 3.6. The authors state that IAE produced a time-dependent reduction in phosphorylation of MEK-1, ERK1/2, and Akt however this may not be an accurate description of what the representative blot in Fig. 6A is showing. Adding a graph showing densitometry and statistical significance may help clarify the trends.
Experiments: How many independent experiments were performed in each assay prior to performing statistical analysis?
How many batches of extract did you make? Were the cells treated with one batch or multiple? Did they all provide the same response?
Figure 1: Statistical analysis data: what do letters on top of bar graphs represent in terms of statistical significance (Fig 1A, 1C, 1D)? please add explanation in figure description
Figure 2E: which phosphorylated Akt residue was examined? Same for all other blots showing phosphorylated proteins. Add the corresponding residues that were examined.
Figure 3A: Make treatment numbers more visible (E.g. adjust font size and/or change colour) as they are representative of treatment numbers in the next two figures. They are not clearly visible and easily blend in with the rest of the figure.
Figure 3D: What do the two different bands (with a significant difference in molecular weight) for C/EBP/ represent? Give explanations
Figure 4A: The quality of the graphs should be improved (resolution). The quality/size of the figures must be increased so the reader can adequately see the shift in cell cycle progression.
Figure 4C and 4E: which mTOR, cdc2, Akt and p70s6K phosphorylated residue is examined?
Figure 5A: which STAT phosphorylated residue is examined?
Figure 6A, B: which MEK1, ERK1/2, Akt p70s6K phosphorylated residue is examined?
Figure 6A and 6B need to be the same size and aligned.
Section 3.7: Line 108: Provide more information on the inhibitors utilized.
Discussion:
Is your in vitro obesity model representative of obesity conditions in adults, children, or both?
It is established that adipocyte number throughout adulthood remains relatively constant and that obesity is attributable to adipocyte hypertrophy. The present in vitro study is just observational. The data show inhibition of lipid accumulation, effects on cell cycle etc with IAE treatment that correlate with inhibition of certain signaling cascades. No data exist to strongly support a cause-effect relationship. In vivo studies are required to provide further valuable information. In light of these, the authors must modify the discussion and conclusion accordingly.
Minor Comments:
Line 36-37: after adipocytes, remove “and” and add a comma. Replace line 37 to “consequently resulting in the development of obesity”
Line 37: remove “of” after “growing”.
Line 40: replace “The major strategies for the improvement of obesity” to “Major strategies to mitigate the disease include”
Line 47” replace “is consisted of” to “consists of”
Line 49-52: Could be reworded for better flow within the sentence.
Lines 55-57: Replace to “Additionally, the activation of extracellular signal-regulated kinase (ERK) 1/2 and Akt signaling cascades is responsible for the MCE, as these signaling pathways mediate cell survival and cell cycle progression, consequently promoting adipogenesis”
Line 58: Remove “Cascades of”
Line 71: Remove “Plant materials, including”
Line 71: Please provide references for the verified safety of dietary polyphenols.
Line 75-77: Please provide references for the listed biological benefits of I. britannica and its bioactive compounds.
Line 86: What antibiotics are in the solution? What are the provided concentrations refering to?
Line 103: “ten times” to be written as 10X
Line 545: Remove “in summary”
Line 138: How much dimethyl sulfoxide was added to each well?
Line 233: perilipin is mentioned and measured before providing any explanation regarding its function. Briefly provide information regarding its function and/or provide why was perilipin measured along with the other adipogenesis markers such as FABP4, FAS, SCD-1.
Line 263: Remove “than that in the positive control.”
Line 284: Should be referencing Fig. S2 not Fig. S1.
Line 284-297: Please provide p-values for theses statistics.
Line 293: Change “decreased by 51.56%” and “increased by 22.87%” to “decreased to 51.56%” and “increased to 22.87%”
Line 316: Please provide a graph showing densitometry for Fig. 4E.
Line 378: Should be referencing Fig. S3 not Fig. S2.
Line 422-423: Please soften this statement. Just because your extract contains compounds that have been shown to be anti-obesogenic in the past, does not mean you can definitively say these compounds are solely responsible for the effect of IAE in your model without performing experiments to show this.
Author Response
Reviewer #1
Comments for Authors:
The study conducted by Hyung-Seok et al. aims to investigate the anti-adipogenic properties of aqueous extract of I. britannica (IAE) on 3T3-L1 preadipocytes and its potential in preventing lipid accumulation. The methodology was extensive and proper, and the data generated point to an anti-obesogenic potential of IAE. It was shown that IAE could effectively downregulate the expression of biomarkers associated with adipogenesis, inhibit cell cycle progression, and regulate the ERK1/2 and Akt signaling pathways. Despite these data, certain points must be addressed as shown below, and extensive proof-reading must be conducted to improve the overall quality of the article.
Major Comments:
The manuscript requires extensive editing. English grammar and syntax are not followed.
→ We appreciated for bringing this weak point to our attention. We revised the whole manuscript, including English grammar and syntax, to improve the quality and correct the errors by editing service.
Line 422-425 “Collectively, IAE suppresses lipid accumulation by inhibiting the development of adipogenesis, which is attributable to its anti-obesogenic phenolic
Line 542: “these findings, our results indicated that the anti-adipogenesis effect of IAE was might be associated with anti-obesogenic phenolic compounds of IAE”
The above statements are not supported by any data. The authors just measured the content/levels of a few polyphenols in their extract. They never tested these polyphenols and therefore the text must be changed to reflect the data accurately.
→ We thank the reviewer for the comment. In advance, we apologize for those impetuous statements. Initially, we meant that these phenolic compounds are might be involved in the adipogenic properties of IAE as various studies demonstrated the anti-obesogenic properties of these phenolic compounds. Now we modified the sentences in revised manuscript to reflect our results appropriately in Line 428-430 and Line 542-543.
Introduction: Include statistics/epidemiology of obesity, risk factors, and provide information regarding the role of adipose tissue in healthy individuals (few sentences).
→ We appreciated for bringing this insufficient points to our attention. We carefully added a few sentences with additional references according to your comments in revised manuscript (Line 32-34, the role of adipose tissue in healthy individuals; Line37-39, risk factors of obesity; Line 39-43, statistics/epidemiology of obesity).
References
- Graham, M.R.; Baker, J.S.; Davies, B. Causes and consequences of obesity: Epigenetics or hypokinesis? Diabetes Meab. Syndr. Obes. Targets Ther. 2015, 8, 455–460.
- Hales, C.; Carroll, M.; Fryar, C.; Ogden, C. Prevalence of obesity and severe obesity among adults: United States, 2017–2018. Available online: https://www.cdc.gov/nchs/products/databriefs/db360.htm.
- Finkelstein, E.A.; Khavjou, O.A.; Thompson, H.; Trogdon, J.G.; Pan, L.; Sherry, B.; Dietz, W. Obesity and severe obesity forcasts through 2030. Am. J. Prev. Med. 2012, 42, 563–570.
Section 2.8 of the methodology: Must verify if results are accurate. How many independent experiments were carried out? How is error accounted for when performing an assay involving quantification following trypsinization? Cells may be lost after washing or depending on time of exposure to trypsin, favouring penetration of trypan blue.
→ We revised the statistical analysis section considering your comments (Line 214-216 in Materials and methods section 2.11.). "The data are presented as mean ± standard deviation of three values obtained from the independent experiment conducted in triplicate at least. The values of respective experiments were determined by mean of three measurements"
As mentioned above, we conducted independent experiments in triplicate and each values of independent experiments were obtained from mean of three measurements. Following this, in case of trypan blue assay, we plated three 6-well plates in same condition as a one batch experiment.
To the best of our knowledge, the trypan blue assay is common methods for counting the viable cells. We agree with your comments that loss of cells is unavoidable during the washing, trypsinization, and staining with trypan blue. However, the loss of the cells was ignored as all experimental procedures are controlled in same conditions when performing the trypan blue assay.
References
- Lee, J.H.; Go, Y.; Lee, B.; Wang, Y.H.; Park, K.I.; Cho, W.K.; Ma, J.Y. The fruits of Gleditsia sinensis Lam. inhibits adipogenesis through modulation of mitotic clonal expansion and STAT3 activation in 3T3-L1 cells. J. Ethnopharmacol. 2018, 222, 61–70.
- Chae, S.Y.; Seo, S.G.; Yang, H.; Yu, J.G.; Suk, S.J.; Jung, E.S.; Ji, H.; Kwon, J.Y.; Lee, H.J.; Lee, K.W. Anti-adipogenic effect of erucin in early stage of adipogenesis by regulating Ras activity in 3T3-L1 preadipocytes. J. Funct. Foods 2015, 19, 700–709.
- Kang, H.J.; Seo, H.A.; Go, Y.; Oh, C.J.; Jeoung, N.H.; Park, K.G.; Lee, I.K. Dimethylfumarate suppresses adipogenic differentiation in 3T3-L1 preadipocytes through inhibition of STAT3 activity. PLoS One, 2013, 8, e61411.
Section 3.5. The authors state that IAE caused a time-dependent reduction in MDI-induced expression of p-STAT3 and C/EPB-β. The representative blots in Fig. 5C and 5F do not adequately reflect this. If IAE time-dependently inhibits the effect of MDI, this should be reflected as a more pronounced reduction in p-STAT3 and C/EPB-β at higher treatment times however this is not what the blots show. Graphs showing densitometry and statistical significance must be added.
→ We thank for bringing this out. In advance, we apologize that we did not added the densitometry graphs of Figs. 5A and 5D. We agree with your comment, however, we just intended to show inhibitory effect of IAE on the phosphorylation of STAT3 and expression of C/EBP-β during early phases of adipogenesis by presenting a time-course phosphorylation of these proteins in the presence or absence of IAE as a preliminary data. Therefore, we re-arranged the order of figure 5, deleted the word "time-dependently" and revised the manuscript in Line 354-362 considering your comments. Please kindly considerate our intends.
Section 3.6. The authors state that IAE produced a time-dependent reduction in phosphorylation of MEK-1, ERK1/2, and Akt however this may not be an accurate description of what the representative blot in Fig. 6A is showing. Adding a graph showing densitometry and statistical significance may help clarify the trends.
→ We thank again for bringing this out and apologize for absence of densitometry graphs of Fig. 6A. We respectfully request your kind consideration on our intends. Same as mentioned above, we presented Fig. 6A to show simple inhibitory effect of IAE on phosphorylation of MEK-1, ERK 1/2 and Akt compared to positive control groups during early phases of adipogenesis presenting a time-course phosphorylation of these proteins with or without IAE treatment as a preliminary data. However, considering your comments, we deleted the word "time-dependently" and revised manuscript in Line 379-387 (in results section 3.6.). And we combined section 3.6. and section 3.7. in revised manuscript considering the other reviewer's comment.
Experiments: How many independent experiments were performed in each assay prior to performing statistical analysis?
→ We revised the statistical section considering your comments (Line 211-213 in Materials and methods section). "The data are presented as mean ± standard deviation of three values obtained from the independent experiment conducted in triplicate at least. The values of respective experiments were determined by mean of three measurements"
How many batches of extract did you make? Were the cells treated with one batch or multiple? Did they all provide the same response?
→ We apologize for the insufficient description. The extracts were independently prepared three times at least. At the screening stage, we confirmed the adipogenesis effect of obtained every extract and they showed similar effects. Following this, an independent experiment was conducted by treating the cells with one batch of extract and we used different batch of extract in next batch of experiment.
Figure 1: Statistical analysis data: what do letters on top of bar graphs represent in terms of statistical significance (Fig 1A, 1C, 1D)? please add explanation in figure description
→ We apologize for confusing statement. Statistical analysis of obtained data was performed and each value were categorized in different groups when p < 0.05 based on one-way analysis of variance, followed by Tukey's post hoc test. Considering your comment, we have revised as "Values labeled with different letters are significantly different (p < 0.05). Differences among the multiple groups were determined based on one-tailed one-way analysis of variance, followed by Tukey's post hoc test" in every figure legend.
Figure 2E: which phosphorylated Akt residue was examined? Same for all other blots showing phosphorylated proteins. Add the corresponding residues that were examined.
→ We have provided the description about phosphorylated residue of Akt that we examined in L101-105 and legend of Fig. 2.
Figure 3A: Make treatment numbers more visible (E.g. adjust font size and/or change colour) as they are representative of treatment numbers in the next two figures. They are not clearly visible and easily blend in with the rest of the figure.
→ We apologize for this inconvenience. We have modified the Fig. 3A by increasing the font size.
Figure 3D: What do the two different bands (with a significant difference in molecular weight) for C/EBP/ represent? Give explanations
→ We thank for pointing this adequacy. We presented description about the isoforms of C/EBP-α in Figs. 2C and 3D.
Figure 4A: The quality of the graphs should be improved (resolution). The quality/size of the figures must be increased so the reader can adequately see the shift in cell cycle progression.
→ We apologize for this inconvenience. We have increased the size of Fig. 4A and improved the resolution (600 dpi).
Figure 4C and 4E: which mTOR, cdc2, Akt and p70s6K phosphorylated residue is examined?
→ We have provided the description about phosphorylated residue of cdc2, mTOR, Akt, and p70S6K that we examined in L101-105 and legend of Fig. 4.
Figure 5A: which STAT phosphorylated residue is examined?
→ We have provided the description about phosphorylated residue of STAT3 that we examined in L101-105 and legend of Fig. 5.
Figure 6A, B: which MEK1, ERK1/2, Akt p70s6K phosphorylated residue is examined?
→ We have provided the description about phosphorylated residue of MEK-1, ERK 1/2, and Akt that we examined in L101-105 and legend of Fig. 6.
Figure 6A and 6B need to be the same size and aligned.
→ We have adjusted size and aligned figures in Fig. 6.
Section 3.7: Line 108: Provide more information on the inhibitors utilized.
→ We have added additional descriptions about inhibitors we used in Line129-130 and Line 391-395.
Discussion:
Is your in vitro obesity model representative of obesity conditions in adults, children, or both?
It is established that adipocyte number throughout adulthood remains relatively constant and that obesity is attributable to adipocyte hypertrophy. The present in vitro study is just observational. The data show inhibition of lipid accumulation, effects on cell cycle etc with IAE treatment that correlate with inhibition of certain signaling cascades. No data exist to strongly support a cause-effect relationship. In vivo studies are required to provide further valuable information. In light of these, the authors must modify the discussion and conclusion accordingly.
→ We appreciate reviewer for pointing this insufficiency out.
Obesity is developed by aberrant expansion of adipose tissue and both of number and size of adipocytes are involved in the growth and expansion of adipose tissue. The development of adipogenesis comprises an increase in number of adipocytes by undergoing MCE (hyperplasia) and enlargement in volume of individual adipocytes by intracellular lipid accumulation (hypertrophy). Accordingly, person who possess high number of adipocytes is vulnerable to obesity compared to who possess low number of adipocytes. Following this, promoted adipogenesis could result in development severe obesity due to increased adipocyte numbers. Therefore, we intended to describe the significance of number of adipocytes not about the models about child and/or adult.
Thus, we have revised the manuscript to appropriately support our intends in Line 437-449.
Minor Comments:
Line 36-37: after adipocytes, remove “and” and add a comma. Replace line 37 to “consequently resulting in the development of obesity”
→ We have revised the sentence according to your comments in Line 37-39.
Line 37: remove “of” after “growing”.
→ We have revised in Line 37.
Line 40: replace “The major strategies for the improvement of obesity” to “Major strategies to mitigate the disease include”
→ We thank reviewer for comments. We have revised in Line 45.
Line 47” replace “is consisted of” to “consists of”
→ We have revised as "comprises" in Line 51.
Line 49-52: Could be reworded for better flow within the sentence.
→ We have modified the structure of text in Line 52-56.
Lines 55-57: Replace to “Additionally, the activation of extracellular signal-regulated kinase (ERK) 1/2 and Akt signaling cascades is responsible for the MCE, as these signaling pathways mediate cell survival and cell cycle progression, consequently promoting adipogenesis”
→ We have revised the sentence as "Additionally, the activation of extracellular signal-regulated kinase (ERK) 1/2 and Akt signaling cascades is responsible for the MCE, as these signaling pathways mediate the cell survival and cell cycle progression, consequently promoting adipogenesis" in Line 59-61.
Line 58: Remove “Cascades of”
→ We have removed "cascades of" in Line 62.
Line 71: Remove “Plant materials, including”
→ We have removed "Plant materials, including" in Line 79.
Line 71: Please provide references for the verified safety of dietary polyphenols.
→ We apologize for inappropriate words upon impetuous referencing. Following your comment, we carefully searched other studies describing the safety of phenolic compounds. And there are some controversies about safety of phenolic compounds. Therefore, we soften the statements with additional references in Line 74-79.
References
- Zhang, H.; Tsao, R. Dietary polyphenols, oxidative stress and antioxidant and anti-inflammatory effects. Curr. Opin. Food Sci. 2016, 8, 33–42.
- Granato, D.; Mocan, A.; Câmara, J.S. Is a higher ingestion of phenolic compounds the best dietary strategy? A scientific opinion on the deleterious effects of polyphenols in vivo. Trends Food Sci. Technol. 2020, 98, 162–166.
Line 75-77: Please provide references for the listed biological benefits of I. britannica and its bioactive compounds.
→ We provided the references, describing biological benefits of I. britannica
- Bae, W.Y.; Kim, H.Y.; Park, E.H.; Kim, K.T.; Paik, H.D. Improved in vitro antioxidant properties and hepatoprotective effects of a fermented Inula britannica extract on ethanol-damaged HepG2 cells. Mol. Biol. Rep. 2019, 46, 6053–6063.
- Kim, H.Y.; Bae, W.Y.; Yu, H.S.; Chang, K.H.; Hong, Y.H.; Lee, N.K.; Paik, H.D. Inula britannica fermented with probiotic Weissella cibaria D30 exhibited anti-inflammatory effect and increased viability in RAW 264.7 cells. Food Sci. Biotechnol. 2020, 29, 569–578.
- Bae, W.Y.; Kim, H.Y.; Kim, K.T.; Paik, H.D. Inhibitory effects of Inula britannica extract fermented by Lactobacillus plantarum KCCM 11613P on coagulase activity and growth of Staphylococcus aureus including methicillin-resistant strains. J. Food Biochem. 2019, 43, e12785.)
and its bio-active compounds
- Bae, W.Y.; Kim, H.Y.; Choi, K.S.; Chang, K.H.; Hong, Y.H.; Eun, J.; Lee, N.K.; Paik, H.D. Investigation of Brassicajuncea, Forsythia suspensa, and Inula britannica: phytochemical properties, antiviral effects, and safety. BMC Complement. Altern. Med. 2019, 19, 253.
- Khan, A.L.; Hussain, J.; Hamayun, M.; Gilani, S.A.; Ahmad, S.; Rehman, G.; Kim, Y.H.; Kang, S.M.; Lee, I.J. Secondary metabolites from Inula britannica L. and their biological activities. Molecules 2010, 15, 1562–1577.
- Cai, Y.; Luo, Q.; Sun, M.; Corke, H. Antioxidant activity and phenolic compounds of 112 traditional Chinese medicinal plants associated with anticancer. Life Sci. 2004, 74, 2157–2184.
- Bai, N.; Zhou, Z.; Znu, N.; Zhang, L. Quan, Z.; He, K.; Zheng, Q.Y.; Ho, C.T. Antioxidative flavonoids from the flower of Inula britannica. J. Food Lipids 2005, 12, 141–149.
- Hong, T.; Zhao, J.; Dong, M.; Meng, Y.; Mu, J.; Yang, Z. Composition and bioactivity of polysaccharides from Inula britannica flower. Int. J. Biol. Macromol. 2012, 51, 550–554.
Line 86: What antibiotics are in the solution? What are the provided concentrations refering to?
→ We apologize for this error. We have revised as " penicillin, 10,000 U/mL; streptomycin, 10,000 μg/mL" in Line 94.
Line 103: “ten times” to be written as 10X
→ We have revised as "subjected to aqueous extraction (1:10; I. britannica powder:distilled water; w/v)" in Line 113-114.
Line 545: Remove “in summary”
→ We have removed "in summary" in Line 554.
Line 138: How much dimethyl sulfoxide was added to each well?
→ We have described added exact volume of dimethyl sulfoxide as "the formazan deposits were dissolved in 1 mL of dimethyl sulfoxide"
Line 233: perilipin is mentioned and measured before providing any explanation regarding its function. Briefly provide information regarding its function and/or provide why was perilipin measured along with the other adipogenesis markers such as FABP4, FAS, SCD-1.
→ We appreciate for pointing this inconsistency out. We have provided the brief information regarding the function of perilipin in Line 72-73.
Line 263: Remove “than that in the positive control.”
→ We have revised the sentence as "Compared with control groups (treatment no. 1), treatment with IAE reduced the intracellular lipid accumulation irrespective of the treatment periods (Figs. 3B and C)." in Line 277-279.
Line 284: Should be referencing Fig. S2 not Fig. S1.
→ We have corrected error as "Fig. S2".
Line 284-297: Please provide p-values for these statistics.
→ Upon your comment, we confirmed the p-value in data of cell cycle progression and the p-value in both of trypan blue assay and FACS analysis lower than 0.001. And different letter indicates significant differences that p-value is lower than 0.05 at least as mentioned above. Thus, we did not present the p-value in both results. We gracefully request your kind consideration on our rebuttal.
Line 293: Change “decreased by 51.56%” and “increased by 22.87%” to “decreased to 51.56%” and “increased to 22.87%”
→ We have revised the sentence " The analysis of cellular DNA content revealed that the proportions of cells at the G0/G1 and S phases in the negative control group were 78.05% and 6.98%, respectively, while those in the positive group were 51.56%, and 22.87%, respectively.".
Line 316: Please provide a graph showing densitometry for Fig. 4E.
→ We have provided the densitometry graphs by addition of Fig. 4F.
Line 378: Should be referencing Fig. S3 not Fig. S2.
→ We have revised as "Fig. 3S".
Line 422-423: Please soften this statement. Just because your extract contains compounds that have been shown to be anti-obesogenic in the past, does not mean you can definitively say these compounds are solely responsible for the effect of IAE in your model without performing experiments to show this.
→ We appreciate for pointing this out. We have revised sentences as "These results suggested that IAE suppresses lipid accumulation by inhibiting the development of adipogenesis, which may be mediated by identified anti-obesogenic phenolic compounds in the extract."

Reviewer 2 Report
In this study, the authors went through an extensive series of experiments to show the effects of IAE on modulating adipogenesis and lipogenesis.
Minor comments:
- Line 74: replace "in" with "is" for "...and in used for medicinal..."
- Line 284: This is Fig. S2 not S1
- Line 304-305: This sentence is confusing, using the word of "improved" is not understood in this context and is a better used in the discussion please use increased to be more clear.
- Line 328: add in the word "adipogenic" to the following sentence "...with alteration in the 'adipogenic' transcription factors,..." inorder to be more specific on what type of transcription factors that you are talking about.
- Line 378: This is Fig. S3 not S2
- Line 542: remove the word "was".
Major comments:
- In Figure 2, the IAE concentrations used in A, C, E do not match the bar graph IAE concentrations given in B, D, F. Please fix or explain this.
- Results section 3.4: the results were not discussed for cdc2, however this complex is talked about in the Discussion (line 488-489), is there a reason you did not state the cdc2 results?
- Results 3.7: I really have a problem with this whole section and Figures S3 and 7. This is all redundant work shown by others. Others have already utilized inhibitors of ERK and Akt to show that the are involved in adipogenesis. So by you utilizing the inhibitors to "verify" that IAE works through these pathways is not true; you are correlating a similar effect but you are not verifying. To trully verify that IAE is modulating adipogenesis through ERK1/2 and Akt then you could utilize an activator of these proteins simultaneously with IAE. The activator should block IAE effects on each part in Figure 7. The IAE data shown in Figure 7 is a repeat of the data that you already have shown in Figure 3B, 3D, 4A, 4C, 5A and then all of the ERK1/2 and Akt inhibitor data is just a repeat of work show and done by others that you have actually sited in this manuscript. Therefore, I think Figures S3 and 7 should be removed and you should replace it with an actual experiment that tries to block IAE action in order to "verify" that IAE modulates adipogenesis through ERK/Akt.
Author Response
Reviewer #2
Comments for Authors:
In this study, the authors went through an extensive series of experiments to show the effects of IAE on modulating adipogenesis and lipogenesis.
Minor comments:
1.Line 74: replace "in" with "is" for "...and in used for medicinal...".
→ Thank you for pointing this out. We have revised as your advice in Line 82.
2.Line 284: This is Fig. S2 not S1.
→ Thank you for bringing this error to our attention. We have corrected the typing mistake in Line 303.
3.Line 304-305: This sentence is confusing, using the word of "improved" is not understood in this context and is a better used in the discussion please use increased to be more clear.
→ We apologize for confusing expression. We carefully revised the sentence considering your advice in Line 322. "significantly mitigated the degradation of p27 (CDK inhibitor 1B)"
4.Line 328: add in the word "adipogenic" to the following sentence "...with alteration in the 'adipogenic' transcription factors,..." in order to be more specific on what type of transcription factors that you are talking about.
→ We appreciated for pointing this out. We have revised the sentence to keep clear as your advice in Line 350-351.
5.Line 378: This is Fig. S3 not S2.
→ We apologize for repetitive typing mistake. We have corrected the error in Line 395.
6.Line 542: remove the word "was".
→ We have revised the sentences according to advice of reviewers in Line 428-430.
Major comments:
- In Figure 2, the IAE concentrations used in A, C, E do not match the bar graph IAE concentrations given in B, D, F. Please fix or explain this.
→ We really apologize for confusion and appreciate for bringing this inconsistency to our attention. We have corrected the error in Figs. 2A, C, and E.
- Results section 3.4: the results were not discussed for cdc2, however this complex is talked about in the Discussion (line 488-489), is there a reason you did not state the cdc2 results?
→ We thank for pointing this inconsistency out. Because IAE-mediated inhibition of cell cycle progression was attributable to cell cycle arrest at the G0/G1 phases, we intended to describe preferentially about proteins mediating the cell cycle transition from the G1 phase to the S phase in results section while we described about overall procedure of cell cycle progression in discussion section.
→ Considering your comment, we added brief description about the results of cyclin B1 and cdc-2 to improve the consistency of manuscript in the revised manuscript in Line 322-324.
" Additionally, down-regulated expression levels of G2/M-mediating proteins (cyclin B1 and p-cdc-2) upon treatment of IAE corroborate the evidence that IAE inhibits the cell cycle progression by arresting the cell cycle at the G0/G1 phase"
- Results 3.7: I really have a problem with this whole section and Figures S3 and 7. This is all redundant work shown by others. Others have already utilized inhibitors of ERK and Akt to show that they are involved in adipogenesis. So by you utilizing the inhibitors to "verify" that IAE works through these pathways is not true; you are correlating a similar effect but you are not verifying. To trully verify that IAE is modulating adipogenesis through ERK1/2 and Akt then you could utilize an activator of these proteins simultaneously with IAE. The activator should block IAE effects on each part in Figure 7. The IAE data shown in Figure 7 is a repeat of the data that you already have shown in Figure 3B, 3D, 4A, 4C, 5A and then all of the ERK1/2 and Akt inhibitor data is just a repeat of work show and done by others that you have actually sited in this manuscript. Therefore, I think Figures S3 and 7 should be removed and you should replace it with an actual experiment that tries to block IAE action in order to "verify" that IAE modulates adipogenesis through ERK/Akt.
→ We appreciate the reviewer for insightful comments.
We fully agree with your comments that utilization of ERK/Akt activator is required to verify our statement "IAE modulates adipogenesis through ERK/Akt". However, due to the time and resource limitations, we could not perform the additional experiments using ERK/Akt activators as you commented. Initially, we intended to simply confirm the effect of blockage of ERK1/2 and Akt on adipogenesis by using specific inhibitors (U0126 and LY294002) to compare with the result of IAE and support the evidence that IAE-mediated inhibition of ERK/Akt is involved in adipogenic effect of IAE as treatment with IAE showed mitigated phosphorylation of MEK-1, ERK 1/2, and Akt.
→ Considering your comment, inappropriate word "verify" have replaced by "confirm" and manuscript has revised by combining the section 3.6. and 3.7., re-arranging the figures (Fig. 7A→Fig. S3B; Fig. 7B→Fig. 6D; Fig. 7C→Fig. S3C; Fig. 7D→removed; Fig. 7E→Fig. 6E), and modifying the sentences. We respectfully request your consideration on our intends.
We are really appreciated for your valuable comments on our manuscript. We have revised and improved the manuscript following your comments and submit the revised manuscript.

Round 2
Reviewer 2 Report
I have no more comments or suggestions.